# China's Green Total Factor Energy Efficiency Assessment Based on Coordinated Reduction in Pollution and Carbon Emission: From the 11th to the 13th Five-Year Plan

Zebin Zheng *, Wenjun Xiao and Ziye Cheng

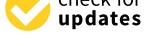

School of Environment and Energy, South China University of Technology, Guangzhou 510006, China; 202121048718@mail.scut.edu.cn (W.X.); 202221047732@mail.scut.edu.cn (Z.C.)
* Correspondence: 202021047885@mail.scut.edu.cn

**Abstract:** As the world's largest energy consumer, China is facing multiple pressures to conserve energy, mitigate pollution and reduce carbon emissions. An objective and scientific assessment of green total factor energy efficiency (GTFEE) is an important prerequisite for achieving energy conservation, emission reduction and low carbon development. In this research, a non-radial data envelopment analysis (DEA) is conducted to assess the GTFEE in China, while the issue of climate and air pollution from energy use is considered in the assessment system. We use different indicators to refer to air pollutants, which provides a reference for related research in indicator selection. The results show that different indicators have different inefficiency values, but changes in the indicators have a minimal effect on the inefficiency values of the other constant indicators. We also assess the GTFEE for the last three five-year plan periods to inform future policy development. The results show that China's average annual GTFEE exhibits a unique trend in each five-year plan period, with an overall "v" shaped trend. The annual average GTFEE of different regions also varies. The other changes in the rankings of the provinces are mainly concentrated in the 11th to 12th Five-Year Plan period.

**Keywords:** green total factor energy efficiency; air pollutant indicators; NDDF; China Five-Year Plan; DEA; GML index



## 1. Introduction

Energy scarcity, severe air pollution, and global climate change are important challenges confronting many developing countries today [1], and atmospheric pollutants and greenhouse gas emissions, which contribute to global climate change, are largely caused by the same process: energy use. China has taken extensive actions in recent years to improve air quality and reduce greenhouse gas emissions, but it remains under dual pressure. On the one hand, China produces a massive amount of air pollution and greenhouse gas emissions. China's total end-use energy consumption accounted for 20.97% of the global consumption in 2019 (China Energy Statistical Yearbook, 2021). As the country with the highest total energy consumption, China emits billions of tons of $CO_2$ [2] and tens of millions of tons of air pollutant emissions each year. On the other hand, as an emerging economics, China continues to rely on economic development to raise the living standards of its citizens. Under dual pressure, policymakers and implementation authorities are focusing on identifying priority governance measures with low social costs. Currently, many studies are in agreement that improving energy efficiency is the key pathway for balancing socioeconomic development and pollution and carbon reduction goals and that the reduction potential and combined benefits of this measure are high [3].

The Chinese government has taken numerous measures to promote energy conservation, energy efficiency improvement, pollutant emission control and $CO_2$ emission intensity reduction since the 11th Five-Year Plan. In terms of energy conservation and energy efficiency improvement, China first proposed the binding target of reducing energy consumption

per unit of gross domestic product (GDP) in the 11th Five-Year Plan (2006–2010) and has maintained it to date. The Chinese government began to call for increasing the share of non-fossil energy in primary energy consumption in the 12th Five-Year Plan (2011–2015). In terms of air pollutant emission control, China listed sulfur dioxide ($SO_2$) as a major pollutant in the 11th Five-Year Plan, requiring a reduction in its total emissions; by the 12th Five-Year Plan, nitrogen oxides ($NO_X$) were added to the major pollutants; and by the 13th Five-Year Plan (2016–2020), China also listed the air quality of prefecture-level cities as a binding indicator, and for cities that do not meet the standard, they are required to decrease their particulate matter $(PM)_{2.5}$ concentration by 18%. In terms of carbon dioxide ($CO_2$) emissions intensity, China first proposed and has maintained a binding target to reduce $CO_2$ emissions per unit of GDP in the 12th Five-Year Plan. China has long proposed a series of plans and policies aimed at reducing energy intensity, $CO_2$ emissions, and air pollutant emissions to achieve sustainable development and build a resource-saving and environmentally friendly society by improving energy efficiency.

"High-quality development" is the product of further upgrading and evolution of the concept of sustainable development, as opposed to low-quality, poorly structured development, which requires economic development processes to intensively use production factor inputs, reduce ecological and environmental costs, and achieve higher levels of socio-economic benefits. Green total factor energy efficiency (GTFEE) measures total factor energy efficiency, which integrates all inputs and outputs, and measures productivity via the relative change between energy inputs and other outputs, which comprehensively reflects the quality of development. Therefore, this paper uses GTFEE to evaluate past and present energy efficiency. Assessing energy efficiency enables us to understand the trends in energy efficiency and the differences among different regions and provides a favorable reference for improving energy efficiency and ultimately achieving sustainable development goals. In production activities, energy is only one of the input factors, and therefore other factors of production besides energy need to be taken into account when assessing the energy efficiency of the whole system. Similar to the research of other scholars, this paper assesses energy efficiency from a total factor productivity perspective and considers both $CO_2$ and air pollutant emissions from the perspective of energy use [4–7]. In addition, many scholars assess GTFEE considering contraction or expansion of all inputs and outputs. The efficiency measured in this case is the combined efficiency of the entire economic production activity, not energy efficiency. This paper refers to the research of Zhou et al. [8] and Wang et al. [9] and defines GTFEE as the relative ratio between the input side, energy, and the output side, which is a combination of $CO_2$ emissions, air pollutant emissions, and economic output, given the given capital and labor inputs. The purpose of this paper is to use GTFEE to assess the spatial and temporal characteristics of China's green development performance at the provincial level from the 11th to 13th Five-Year Plan period. Furthermore, to emphasize the connotation of energy efficiency, the non-radial directional distance function (NDDF) model of the data envelopment analysis (DEA) model is chosen for assessment. DEA is a widely used method for assessing GTFEE [10–21]. This method does not require pre-assumptions about the form of the production function and avoids the risk of incorrectly setting the function form [22]. Furthermore, the DEA model setting is flexible and versatile, as is indicator selection, allowing it to meet the research needs of various researchers. In terms of the scope of the analysis, DEA has the capability to assess efficiency at various levels including cities [9], provinces [23], regions [14,24], and at even the national level [25]. NDDF models are frequently employed in DEA to determine other efficiencies, such as carbon emission performance [8,26,27]. Efficiency assessments may be on the high side [28] by non-zero slack when using Chambers' [29] directional distance function measure of efficiency. In contrast, the NDDF model developed by Zhou et al. [8] allows inputs and outputs to be adjusted in different proportions. This advantage enables us to avoid the consideration of contraction or expansion of all inputs and outputs (ensuring that the inputs of capital and labor remain constant) and ensures that we obtain the energy efficiency rather than the combined efficiency of the whole system. On the other hand,

we can also use this method to obtain the potential for improvement in each input and output, identify the inefficiencies of the subject, and provide more targeted suggestions for improving efficiency.

This paper also attempts to conduct a comparative study in terms of indicator selection when measuring GTFEE. The selection of assessment indicators in previous studies on total factor energy efficiency was relatively simple. Hu et al. [20] and Chang et al. [30] assessed energy efficiency in 29 provincial administrative regions in early China using the Charnes–Cooper–Rhodes (CCR)-DEA [31] model with GDP as the only output. Similarly, Honma et al. [18] calculated total factor energy efficiency for 14 developed countries using GDP as the only output from 1995 to 2005. As the study progressed, Yeh et al. [21] added $CO_2$ and $SO_2$ emissions as undesirable outputs and utilized the Banker–Charnes–Cooper (BCC)-DEA [32] model to assess total factor energy efficiency in 30 Chinese provinces from 2002 to 2007. According to the findings, the eastern region had the highest energy efficiency, while the western region exhibited poor energy efficiency. Wang et al. [14] also used $CO_2$ and $SO_2$ as undesirable outputs to assess the energy efficiency of China's provinces between 2000 and 2008. In a recent study, Sueyoshi et al. [33] included three indicators, $SO_2$, $NO_X$, and $CO_2$, as undesirable outputs in the energy efficiency assessment system. Qin et al. [19] employed an epsilon-based measure (EBM)-DEA [34] model to analyze the energy efficiency of China's coastal areas from 2000 to 2012, simultaneously focusing on multiple air pollutant emissions and including $NO_X$ as undesirable outputs of energy use in addition to $CO_2$ and $SO_2$. The results show that the energy efficiency scores of most provinces decrease when multiple undesirable outputs are considered. A synthesis of the existing literature shows that scholars are increasingly using non-radial slacks-based measure (SBM), EBM or NDDF models in their methods, instead of radial CCR or BCC models. In early research on the selection of indicators, scholars neither paid attention to the environmental impacts of energy use nor considered the undesirable outputs. As the research progressed, $CO_2$ and $SO_2$ were slowly being incorporated into the assessment system to characterize the problems of climate change and air pollution caused by energy use. In terms of air pollution, with the government's attention to environmental issues, an increasing number of pollutants are classified as major pollutants and subject to strict emission restrictions. Therefore, more major air pollutants need to be considered when selecting indicators that refer to air pollution problems. In addition, there is a lack of comparative research on the differences in energy efficiency assessment results due to different undesirable outputs choices. In this study, on the one hand, we include three current major air pollutants in the assessment system; on the other hand, this paper also hopes to improve the reliability of the results by comparing the measurement results of different indicators and to more fully characterize the spatial and temporal characteristics of green development performance between the 11th and 13th Five-Year Plan periods.

This paper contributes to existing research in two ways: first, in terms of methodology, this paper chooses to measure GTFEE while maintaining non-energy inputs using the NDDF model. In contrast to the traditional DEA model based on the radial distance function, the NDDF model makes specific suggestions for energy use and helps us explore more targeted energy adjustment strategies in energy management and regulation. Second, the selection of undesirable outputs in this paper is achieved using a variety of options. A more reliable assessment of provincial GTFEE is made possible by contrasting various indicator systems. The comparative analysis of indicator selection may also be utilized to assess total factor productivity based on pollution and carbon reduction, offering fresh perspectives on how to choose or create undesirable outputs.

The remainder of this paper is organized as follows: Section 2 describes the methods and data. Section 3 presents the related results and discussions, and Section 4 concludes the paper.

## 2. Methods and Data

### 2.1. DEA Model Based on the NDDF

This paper uses the NDDF DEA model to assess GTFEE. First, this paper supposes a production possibility set containing a single desirable output and two undesirable outputs for the efficiency assessment of N Chinese provinces. There are N decision making units (DMUs). It is assumed that each DMU uses energy ($E$), labor force ($L$), and capital stock ($K$) as inputs during the production process to generate the desirable economic output of GDP ($Y$) while producing undesirable outputs, such as $CO_2$ ($C$) and air pollution ($AP$). The production possibility set ($T$) is defined as:

$$T = \{(K, L, E, Y, AP, C) : (K, L, E) \text{ can produce } (Y, AP, C)\} \tag{1}$$

According to Faere [35], the T must satisfy the weak disposability assumption (I) and null-jointness assumption (II), which are denoted as:

(I) $If(K, L, E, Y, AP, C) \in T$ and $0 < \theta \leq 1$, then $(K, L, E, \theta Y, \theta AP, \theta C) \in T$.

(II) $If(K, L, E, Y, AP, C) \in T$ and $AP = C = 0$, then $Y = 0$.

Assumption I implies that to reduce the undesirable output, one must also reduce the desirable output, that is, reducing the undesirable output has a cost. Assumption II implies that there is no desirable output without undesirable output. $AP$ and $CO_2$ emissions are unavoidable during the production process. Since $T$ is only conceptually defined and lacks a concrete form, it cannot be directly applied in empirical studies. A common solution is to formulate them within a nonparametric piecewise linear framework (DEA approach). $K_n$, $L_n$, $E_n$, $Y_n$, $AP_n$, and $C_n$ are vectors representing capital inputs, labor force inputs, energy inputs, GDP output, air pollutant emissions and $CO_2$ emissions, respectively. Then, $T$ exhibiting constant returns to scale is represented by

$$T = \{(K, L, E, Y, AP, C) : \sum_{n=1}^{N} z_n K_n \leq K$$
$$\sum_{n=1}^{N} z_n L_n \leq L$$
$$\sum_{n=1}^{N} z_n E_n \leq E$$
$$\sum_{n=1}^{N} z_n Y_n \geq Y$$
$$\sum_{n=1}^{N} z_n AP_n = AP$$
$$\sum_{n=1}^{N} z_n C_n = C$$
$$z_n \geq 0, n = 1, 2, \ldots, N\} \tag{2}$$

where $z_n$ is the intensity variable used to construct $T$ by convex combination. Once the $T$ set is well-constructed, the non-radial directional distance functions can be used to assess the GTFEE. Referring to the research method of Zhou et al. [8] on the non-radial directional distance function, this paper defines the non-radial directional distance function as follows:

$$\overrightarrow{D}(K, L, E, Y, AP, C; g) = \sup\{w^T \beta : ((K, L, E, Y, AP, C) + g \times diag(\beta)) \in T\} \tag{3}$$

where $w = (w_K, w_L, w_E, w_Y, w_{AP}, w_C)^T$ is the weight vector of inputs and outputs, $g = (-g_K, -g_L, -g_E, g_Y, -g_{AP}, -g_C)$ is the explicit directional vector, the symbol diag means the diagonal matrices, and $\beta = (\beta_K, \beta_L, \beta_E, \beta_Y, \beta_{AP}, \beta_C)^T \geq 0$ is the vector of the scaling factors representing the individual inefficiency measure for each input and output. Because the focus of this paper is on the potentially adjustable amount of energy inputs, the non-energy inputs ($K$ and $L$) are fixed in the model, and the direction vector is set as $g = (0, 0, -E, Y, -AP, -C)$. In accordance with common approaches in the literature [8,9,36–38],

this paper assigns the same weights to the inputs, desirable outputs, and undesirable outputs, as well as *AP* and *C* in the undesirable outputs, and the final weight vector is set to $w = (0, 0, \frac{1}{3}, \frac{1}{3}, \frac{1}{6}, \frac{1}{6})^T$. Therefore, the value of Equation (3) $\vec{D}(K, L, E, Y, AP, C; g)$ is solved by the following DEA model:

$$
\begin{aligned}
\vec{D}(K, L, E, Y, AP, C; g) = \max & w_E \beta_E + w_Y \beta_Y + w_P \beta_{AP} + w_C \beta_C \\
s.t. \sum_{n=1}^{N} & z_n K_n \leq K \\
\sum_{n=1}^{N} & z_n L_n \leq L \\
\sum_{n=1}^{N} & z_n E_n \leq E - \beta_E g_E \\
\sum_{n=1}^{N} & z_n Y_n \geq Y + \beta_Y g_Y \\
\sum_{n=1}^{N} & z_n AP_n = AP - \beta_{AP} g_{AP} \\
\sum_{n=1}^{N} & z_n C_n = C - \beta_C g_C \\
& z_n \geq 0, n = 1, 2, \ldots, N \\
& \beta_E, \beta_Y, \beta_{AP}, \beta_C \geq 0
\end{aligned}
\tag{4}
$$

where $\beta$ indicates the degree of inefficiency of the variables, that is, the proportion of each variable that can be improved. For the inputs and undesirable outputs, $\beta$ implies the proportion that can be reduced. For the desirable output, $\beta$ implies the proportion that can be increased. Note $\beta$ is calculated from the model and may not be fully realized in practice. A larger $\beta$ indicates a larger proportion of that variable that can be improved and lower efficiency, while the opposite indicates a higher efficiency. The analysis of $\beta$ explains why GTFEE is at a low level and facilitates the proposal of more targeted policies. If $\beta$ is equal to zero for all variables, then $\vec{D}(K, L, E, Y, AP, C; g) = 0$. Thus, there is no room for improvement in all variables of this DMU. The DMU is located on the production frontier. Supposing that $\beta^*$ is the optimal solution of the DEA model, then GTFEE is expressed as:

$$
\text{GTFEE} = \frac{\frac{1}{3}\left[(1 - \beta_E^*) + (1 - \beta_{AP}^*) + (1 - \beta_C^*)\right]}{1 + \beta_Y^*}
\tag{5}
$$

The GTFEE falls between zero and unity. A higher GTFEE indicates that the DMU is able to produce more regional GDP with less energy while emitting fewer pollutants and $CO_2$.

### 2.2. Global Malmquist–Luenberger Index

The Malmquist productivity index is a classic indicator used to measure a change in productivity [39]. As the study evolved, researchers discovered that the Malmquist productivity index was unable to account for undesirable outputs when measuring efficiency changes in multi-input, multi-output models. Therefore, Chung et al. [40] proposed a Malmquist index containing undesired output, named the Malmquist–Luenberger (ML) index. The ML index is now widely utilized to assess the change in efficiency containing undesirable output. However, ML index only compares the efficiency change of two adjacent periods, cannot compare the change between two different periods, and cannot fully reflect the change in consecutive periods, and also may have no solution. Based on the ML index, Pastor and Lovell [41] proposed the Global ML (GML) index. In contrast to the ML index, the GML index uses the technology frontier, which is jointly constructed for all periods, as the reference frontier. Using the global frontier, GML index is able to show both efficiency changes in consecutive periods and those in different periods. Simultaneously, GML solves the problem that ML may be unresolved.

In this paper, we use the *GML* index to analyze the time variation of *GTFEE*, which is shown in Equations (6)–(8). The superscript G of *GTFEE* indicates that the frontier used for this efficiency value is composed of DMUs from all periods; the superscript $t$ or $t-1$ indicates that the frontier is composed of DMUs from the current period (period $t$ or period $t-1$, respectively) only. The input-output indicator's superscripts $t$ and $t-1$ indicate that the input-output is from period $t$ or $t-1$, respectively; the subscript $n$ indicates the $n$th DMU.

$$GML_n(t-1,t) = \frac{GTFEE_n^G\left(K_n^t, L_n^t, E_n^t, Y_n^t, P_n^t, C_n^t\right)}{GTFEE_n^G\left(K_n^{t-1}, L_n^{t-1}, E_n^{t-1}, Y_n^{t-1}, P_n^{t-1}, C_n^{t-1}\right)} \tag{6}$$

$GML_n(t-1,t)$ is used to assess the overall change in the GTFEE of the $n$th DMU from year $t-1$ to year $t$. If $GML_n(t-1,t) > 1$, GTFEE increases; if $GML_n(t-1,t) < 1$, GTFEE decreases. The GML index can be further decomposed into two components: technical efficiency change (EC) and technological change (TC) in the production technology of the DMU between two periods, EC and TC are expressed as:

$$EC_n(t-1,t) = \frac{GTFEE_n^t\left(K_n^t, L_n^t, E_n^t, Y_n^t, P_n^t, C_n^t\right)}{GTFEE_n^{t-1}\left(K_n^{t-1}, L_n^{t-1}, E_n^{t-1}, Y_n^{t-1}, P_n^{t-1}, C_n^{t-1}\right)} \tag{7}$$

$$TC_n(t-1,t) = \frac{\frac{GTFEE_n^G\left(K_n^t, L_n^t, E_n^t, Y_n^t, P_n^t, C_n^t\right)}{GTFEE_n^t\left(K_n^t, L_n^t, E_n^t, Y_n^t, P_n^t, C_n^t\right)}}{\frac{GTFEE_n^G\left(K_n^{t-1}, L_n^{t-1}, E_n^{t-1}, Y_n^{t-1}, P_n^{t-1}, C_n^{t-1}\right)}{GTFEE_n^{t-1}\left(K_n^{t-1}, L_n^{t-1}, E_n^{t-1}, Y_n^{t-1}, P_n^{t-1}, C_n^{t-1}\right)}} \tag{8}$$

EC indicates the relative rate of change of technical efficiency between two periods. If EC > 1, the technical efficiency has improved, which indicates that DMU is closer to the frontier side of the same period compared with the previous period and has a tendency to catch up with the frontier. Otherwise, the technical efficiency decreases and moves away from the frontier. For TC, the numerator in Equation (8) reflects the distance between the $t$ period frontier of the $n$th DMU and the global frontier, and a larger value indicates that the $t$ period frontier is closer to the global frontier. Similarly, the denominator reflects the distance between the $t-1$ period frontier of the $n$th DMU and the global frontier. Therefore, TC measures the change in the technology frontier of the nth DMU with the change in time. TC > 1 indicates technological progress and TC < 1 indicates technological regression.

### 2.3. Indicators and Data

The 11th to 13th Five-Year Plan period covers three five-year plans in China. During this period, China's economy rapidly developed, with the per capita GDP increasing from 16,700 RMB (2006, current year prices) to 71,800 RMB (2020, current year prices). Simultaneously, national and local governments in China implemented large-scale and comprehensive environmental management under immense pressure on the ecological environment. Therefore, the period selected for the energy efficiency assessment in this paper is from the 11th Five-Year Plan to the 13th Five-Year Plan. Considering the large change in data in 2020 due to the Coronavirus Disease 2019 (COVID-19) pandemic, 2006-2019 was ultimately chosen as the study period. The study population comprised 30 provinces in China (Tibet, Hong Kong, Macau, and Taiwan are not included in the assessment due to missing data). The three input indicators employed in assessing GTFEE are $K$, $L$, and $E$; the desirable output is $Y$, and there are two types of undesirable outputs, $C$ and $AP$, including: $SO_2$, $NO_X$, PM, and pollutant equivalents.

#### 2.3.1. Air Pollutant Indicators

In this paper, three single indicators ($SO_2$ emissions, $NO_X$ emissions, and PM emissions) and one composite indicator are used to refer to air pollutants. The composite indicator was calculated by referring to the research method of Mao et al. [42–44] in the field of synergistic control, using the equivalent value of air pollutants (the equivalent

value reflects the degree of harm of pollutants and their treatment costs) and pollutant emissions in the Environmental Protection Tax Law of the People's Republic of China. The composite indicator was calculated as follows:

$$P = \sum \alpha_i \cdot p_i \tag{9}$$

where $P$ is the air pollution equivalent value, which is the composite indicator used in this paper; $\alpha_i$ is the conversion coefficient of the $i$th pollutant, and the specific values are shown in Table 1; and $p_i$ is the actual emission of the ith pollutant (data from the China Statistical Yearbook and the China Environment Yearbook). This composite indicator has been widely employed in studies on the synergistic control effects of air pollutants and greenhouse gases in China [45–50] but has not been applied to the analysis and assessment of energy efficiency.

**Table 1.** Air pollutant conversion coefficient.

| Air Pollutants | Conversion Coefficient | Value |
|:---:|:---:|:---:|
| $SO_2$ | $\alpha_1$ | $1/(0.95\,^1)$ |
| $NO_X$ | $\alpha_2$ | $1/(0.95\,^1)$ |
| PM | $\alpha_3$ | $1/(2.18\,^1)$ |

[1] The value is derived from the Environmental Protection Tax Law of the People's Republic of China.

In this paper, there are four metrics that refer to air pollutants in the undesirable output: $SO_2$ emissions (S), $NO_X$ emissions (N), PM emissions (PM), and air pollutant equivalents (P). In the subsequent results and discussion, this paper uses different subscripts to indicate the air pollutant indicators used for GTFEE. See Table 2 for details.

**Table 2.** Inputs and outputs of the four types of GTFEE.

| GTFEE | Inputs | | | Desirable Output | Undesirable Outputs | |
|:---:|:---:|:---:|:---:|:---:|:---:|:---:|
| | | | | | Greenhouse Gas | Air Pollutant |
| $GTFEE_P$ | Capital | Labor force | Energy | GDP | $CO_2$ emissions | Air pollution equivalent value |
| $GTFEE_S$ | Capital | Labor force | Energy | GDP | $CO_2$ emissions | $SO_2$ emissions |
| $GTFEE_N$ | Capital | Labor force | Energy | GDP | $CO_2$ emissions | $NO_X$ emissions |
| $GTFEE_{PM}$ | Capital | Labor force | Energy | GDP | $CO_2$ emissions | PM emissions |

### 2.3.2. $CO_2$ Emission Indicators

Global climate change is an important issue for many countries, and excessive $CO_2$ emissions are one of the main causes of climate change. The $CO_2$ emissions data in this paper are obtained from the Carbon Emission Accounts and Datasets (CEDAs) [2,4,51,52], which is calculated by the Intergovernmental Panel on Climate Change (IPCC) sectoral accounting method. According to this data source, the total $CO_2$ emissions of 30 Chinese provinces grew from 6198 Mt in 2006 to 10,882 Mt in 2019, ranking among the top worldwide.

### 2.3.3. Other Input-Output Indicators

The $K$ is estimated by referring to the method of Shan [53], using the perpetual inventory method, and the data base is the social fixed asset investment data from the China Statistical Yearbook; $L$ is taken as the number of employed people in the whole society, and the data is obtained from the data for each province's statistical yearbook; $E$ is the comprehensive energy consumption from the China Energy Statistical Yearbook; and $Y$ is each province's GDP from the China Statistical Yearbook. The descriptive statistics of all input-output indicators are shown in Table 3.

**Table 3.** Descriptive statistics of input and output indicators for four years.

| Inputs and Outputs | Year | 2006 | 2010 | 2015 | 2019 |
|---|---|---|---|---|---|
| Capital | Mean | 1364.92 | 2921.85 | 6497.86 | 9894.93 |
| (billion RMB) | Std. Dev. | 1046.70 | 1982.08 | 4093.22 | 6453.72 |
| | Max | 3901.45 | 8027.84 | 17,276.27 | 25,874.06 |
| | Min | 105.27 | 310.89 | 918.80 | 1531.73 |
| Labor force | Mean | 24.18 | 25.85 | 27.66 | 27.50 |
| (million people) | Std. Dev. | 16.49 | 17.51 | 18.27 | 18.12 |
| | Max | 59.60 | 64.02 | 66.42 | 71.50 |
| | Min | 2.94 | 3.08 | 3.21 | 3.30 |
| Energy | Mean | 96.85 | 129.84 | 146.92 | 163.11 |
| (million tce) | Std. Dev. | 62.73 | 81.72 | 89.39 | 96.32 |
| | Max | 267.59 | 348.08 | 393.32 | 413.90 |
| | Min | 9.20 | 13.59 | 19.16 | 22.64 |
| GDP | Mean | 735.63 | 1162.98 | 1816.65 | 2380.64 |
| (billion RMB) | Std. Dev. | 610.63 | 954.87 | 1458.49 | 1916.58 |
| | Max | 2596.12 | 4078.83 | 6149.18 | 8059.91 |
| | Min | 58.52 | 92.39 | 151.31 | 198.70 |
| Pollutant emissions | Mean | 1732.29 | 1611.94 | 1534.68 | 756.20 |
| (kilotons of pollutant equivalents) | Std. Dev. | 1012.97 | 889.67 | 898.70 | 435.56 |
| | Max | 3793.95 | 3367.36 | 3601.40 | 1617.62 |
| | Min | 97.00 | 96.14 | 137.57 | 69.12 |
| $CO_2$ | Mean | 206.60 | 286.62 | 327.72 | 362.72 |
| (million $tCO_2$) | Std. Dev. | 141.57 | 191.00 | 210.29 | 243.85 |
| | Max | 605.51 | 795.49 | 854.46 | 937.12 |
| | Min | 19.19 | 28.93 | 42.28 | 43.07 |
| $SO_2$ | Mean | 862.20 | 728.26 | 619.53 | 152.32 |
| (kiloton) | Std. Dev. | 510.46 | 411.50 | 363.32 | 94.74 |
| | Max | 1962.00 | 1537.80 | 1525.70 | 352.40 |
| | Min | 24.00 | 28.80 | 32.30 | 1.90 |
| $NO_X$ | Mean | 507.93 | 617.50 | 615.25 | 409.92 |
| (kiloton) | Std. Dev. | 357.84 | 390.20 | 363.76 | 268.95 |
| | Max | 1247.00 | 1408.00 | 1423.90 | 1093.30 |
| | Min | 59.00 | 56.00 | 89.50 | 48.70 |
| PM | Mean | 632.30 | 425.87 | 512.10 | 358.32 |
| (kiloton) | Std. Dev. | 426.48 | 249.17 | 379.05 | 226.50 |
| | Max | 1704.00 | 986.00 | 1575.40 | 957.60 |
| | Min | 21.00 | 15.00 | 20.40 | 15.40 |

## 3. Results and Discussions

We expect to compare a variety of undesirable output indicators to more robustly assess China's provincial GTFEE, and to assess the change in China's provincial GTFEE between the 11th Five-Year Plan period and the 13th Five-Year Plan period based on the NDDF and GML index models. In the first section of this part, we compare the differences in GTFEE assessed by different undesirable output indicators, identify the source of the differences, and analyze the differences by region. A more robust composite indicator is chosen as the undesirable output. Sections 2 and 3 assess the efficiency in terms of the global static GTFEE and the dynamic GML index, respectively. Section 4 analyzes the changes in the ranking of each province in these three five-year plans. This paper explores more targeted strategies in energy management and regulation by analyzing the spatial and temporal characteristics of GTFEE at the provincial level in China over the past three five-year plans in the context of current pollution reduction and carbon reduction.

### 3.1. Undesirable Output Indicator Selection and Comparison

We use four different indicators to separately represent the air pollutants in the undesirable output and asses GTFEE. Figure 1 shows the national annual average values of

the four types of GTFEE during the study period (different subscripts represent the air pollutant indicators for assessing GTFEE). China's GTFEE shows a similar trend regardless of which undesirable output is utilized. GTFEE$_P$ is always in between the 4 types of GTFEE.

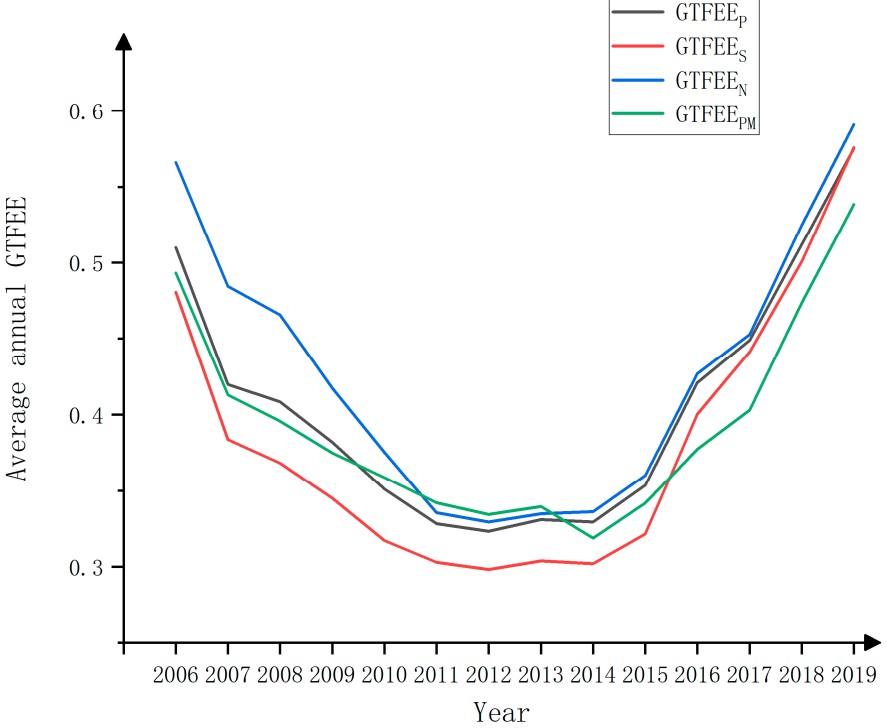

**Figure 1.** Variation in national average annual GTFEE.

Due to the vast territory of China, provinces in different geographic locations have different resources and levels of development, and their development rates and stages vary, as do the problems they face. Thus, generalizations are impossible. This research groups the 30 provinces into eastern, central, and western regions according to their geographical locations [38]. Figure 2 shows the comparison of the four types of GTFEE for the three regions. The results indicate that the four types of GTFEE have similar results regardless of region, and that GTFEE$_P$ is always in an intermediate position.

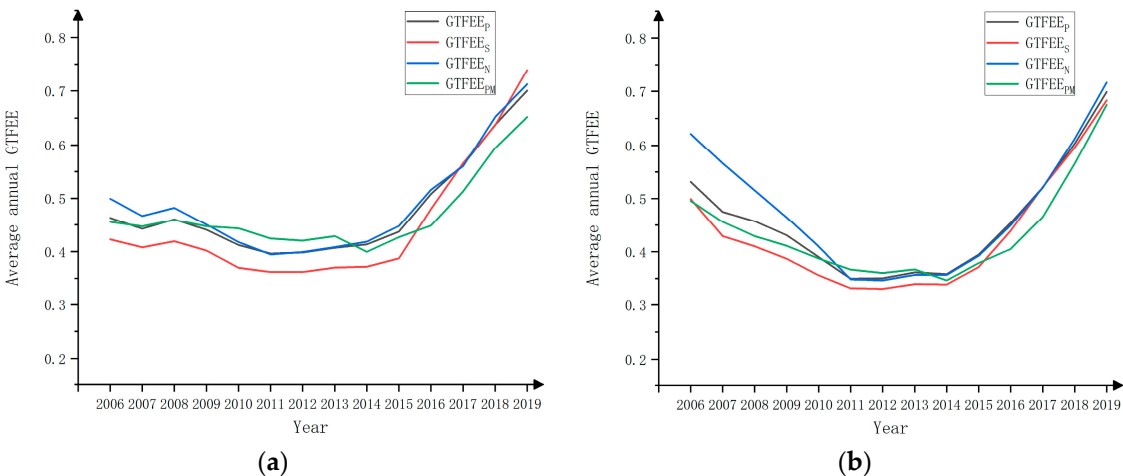

(a)                    (b)

**Figure 2.** *Cont.*

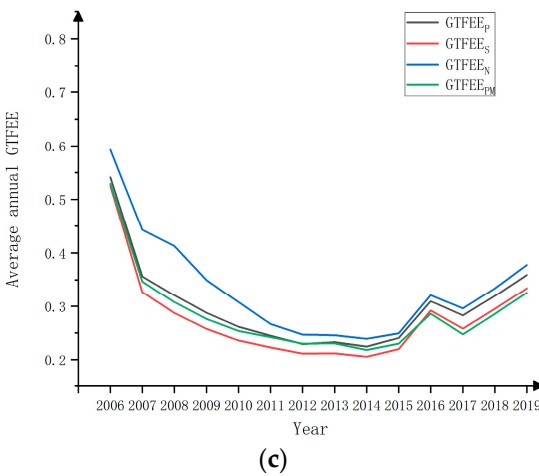

(**c**)

**Figure 2.** Variation in annual average GTFEE in different regions: (**a**) east, (**b**) central, (**c**) west.

To further explore the sources of variation, we analyzed the β value used to calculate GTFEE. The results are shown in Figure 3. The difference in GTFEE when different indicators are used to refer to air pollutants for assessment is mainly derived from the change in the inefficiency of different air pollutant indicators, which has a minimal effect on the β of other indicators. When $NO_X$ is selected as an indicator of air pollutants, its β rises much faster than the other three indicators. When $SO_2$ is utilized as an indicator of air pollutants, its rate of increase is slow, but its inefficiency level is always high (near 0.8) and substantially higher than that of the other indicators.

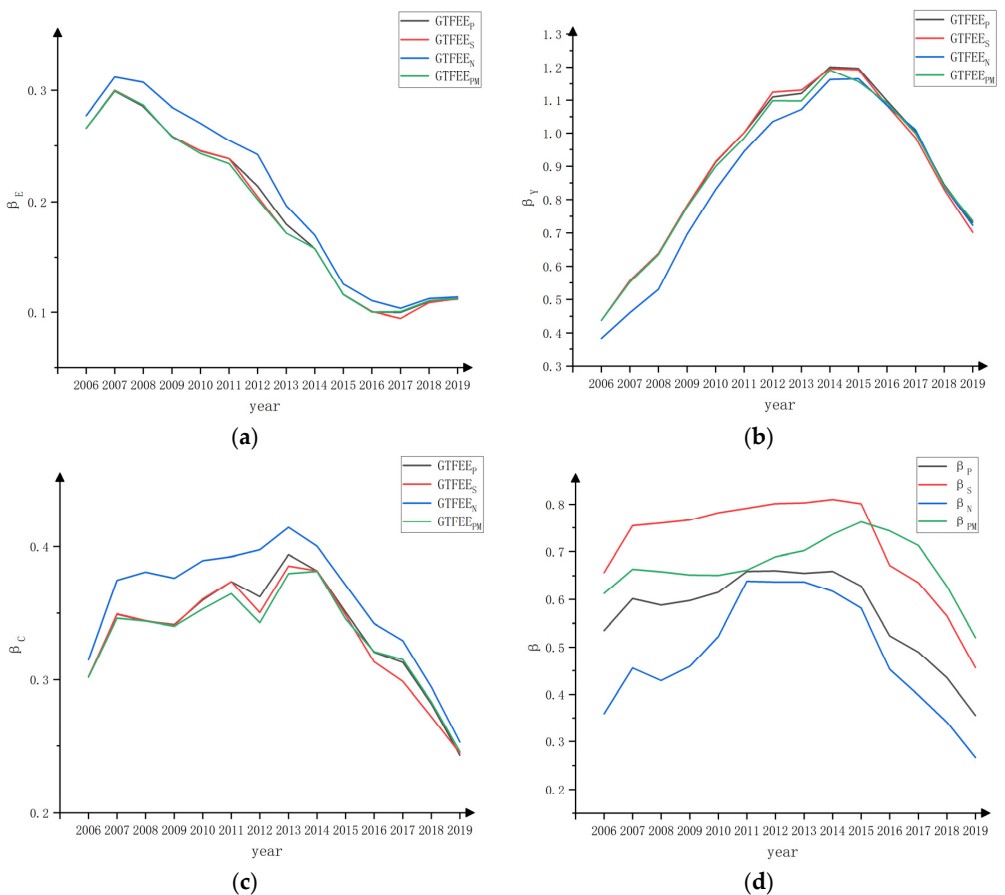

**Figure 3.** Variation in national average annual β: (**a**) input (E), (**b**) desirable output (Y), (**c**) undesirable output (C), (**d**) undesirable output (AP).

Based on the results of the above analysis, we focus on the inefficiency values of air pollutants to further analyze each region; the results are shown in Figure 4. A comparison of Figure 4 with Figure 3d reveals some differences among the regions. $\beta_S$ shows the same trend in all regions. Higher inefficiency values are observed in the eastern and western regions, but by 2019, inefficiency values in the eastern region have declined to a lower level, while those in the western region exhibit a slower decreasing trend. $\beta_N$ is different: the central and western regions are still efficient in 2006 ($\beta_N$ is 0.23 in the central region and 0.29 in the western region), while $\beta_N$ in the eastern region has reached 0.52. During subsequent development, $\beta_N$ in the central and western regions rapidly rises and reaches a maximum of 0.62 and 0.65, respectively, a level similar to that of the eastern region. After 2014, $\beta_N$ rapidly decreases and reaches lower levels in the eastern and central regions, while it decreases at a slower rate in the western region.

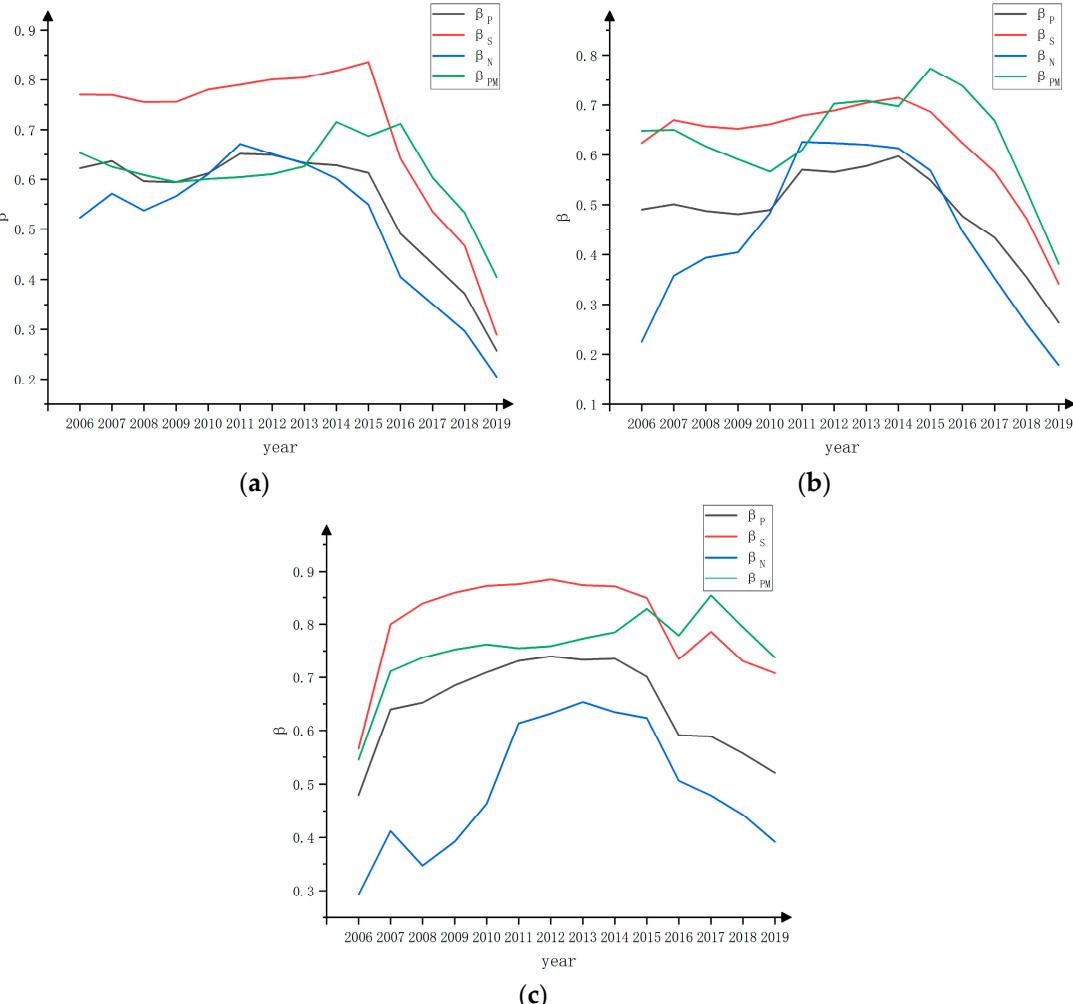

**Figure 4.** Variation in annual average $\beta$ in undesirable output (AP) by region: (**a**) east, (**b**) central, (**c**) west.

This section analyzes the impact of different air pollutant indicators in the undesirable output on the GTFEE assessment and the sources of efficiency differences. The results show that all four types of GTFEE have similar results from both a regional perspective and national perspective, which may be related to China's air pollutant control policies. Policy development tends to focus on the overall atmospheric environment rather than on the control of a single pollutant. Further, this research finds that the differences in the four types of GTFEE mainly arise from the differences in the inefficiency values of the air pollutant indicators. Using different air pollutant indicators has less impact on the inefficiency values of other inputs

and outputs. There are large differences in the inefficiency values of different air pollutant indicators. Using a single indicator to refer to air pollutants may not comprehensively reflect the GTFEE, while using a composite indicator can yield more robust measurement results. Based on the results in this section, we choose GTFEE$_P$ as the target of our analysis in the following analysis.

### 3.2. Global Static GTFEE Analysis

In this section, we follow the results of Section 3.1, that is, we choose the composite indicator (P) to refer to air pollutants to assess the global static GTFEE for China (global static means that the reference frontier of GTFEE is all years that include DMUs). We integrated the national GTFEE$_P$ and three regional GTFEE$_P$ in Figure 5. We discover that the national average annual GTFEE$_P$ are at a low level (highest value of 0.57 and lowest value of 0.32) and show a decreasing trend from 2006–2010, fluctuating at low values from 2011–2014, and starting to increase in 2015. Combined with Figure 3, during the 11th Five-Year Plan period, while the inefficiency of energy inputs is decreasing, the inefficiency of air pollutants and $CO_2$ is still increasing, resulting in a decreasing trend in China's average annual GTFEE$_P$. During the 12th Five-Year Plan period, China paid more attention to green development and the environmental problems caused by development, updating and introducing a series of emission standards for heavily polluting industries (Table 4) and by proposing more stringent environmental development requirements. During this five-year period, the inefficiency level of air pollutants stopped rising, and China's annual average GTFEE$_P$ stopped falling, fluctuating at a low level. With the 13th Five-Year Plan, China included carbon emission intensity reduction as a mandatory target in China's development goals, in addition to the original air pollution control measures. Under the dual pressure on air pollutants and carbon emissions management, the inefficiency of air pollutants and $CO_2$ started to decline and China's annual average GTFEE$_P$ started to rise from 2015. This finding is different from those of previous studies [14,54]. After including $CO_2$ and air pollutants in the assessment system, we discover that the GTFEE$_P$ in China does not show an increasing trend from year to year, but a "v" shaped trend. The inefficiency of $CO_2$ has been increasing during the 11th Five-Year Plan period, probably because $CO_2$ emissions were not emphasized during this period. $CO_2$ emissions are often disregarded in previous studies, thus causing discrepancies between the results of this research and those of previous studies.

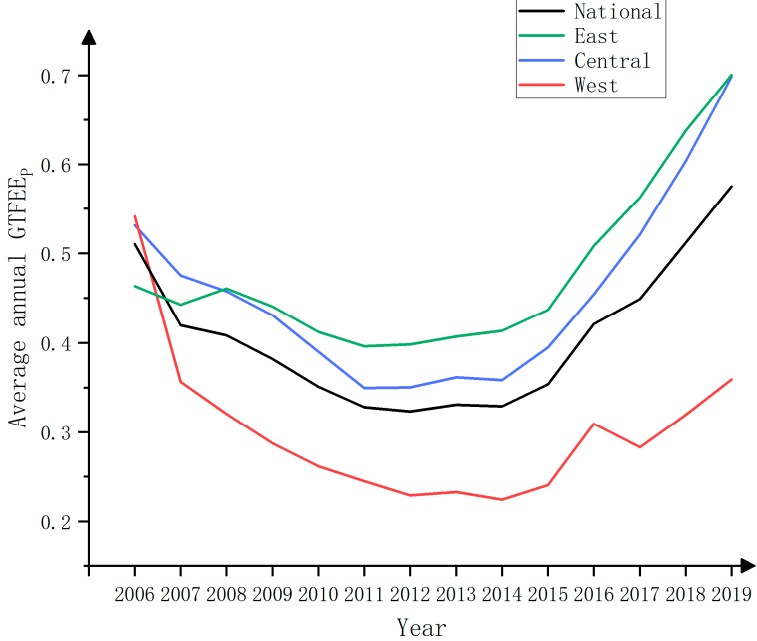

**Figure 5.** Variation in national and regional average annual GTFEE$_P$.

**Table 4.** Some of the new or updated national standards in 2011–2013.

| Industry | Standard Number | Standard Name |
|---|---|---|
| Thermal Power Plant | GB 13223-2011 | Emission standard of air pollutants for thermal power plants [55] |
| Flat Glass | GB 26453-2011 | Emission standard of air pollutants for flat glass industry [56] |
| Sintering | GB 28662-2012 | Emission standard of pollutants for sintering and pelletizing of iron and steel [57] |
| Coking | GB 16171-2012 | Emission standard of pollutants for coking chemical industry [58] |
| Ironmaking | GB 28663-2012 | Emission standard of air pollutants for iron smelt industry [59] |
| Steelmaking | GB 28664-2012 | Emission standard of air pollutants for steel smelt industry [60] |
| Steel Rolling | GB 28665-2012 | Emission standard of air pollutants for steel rolling industry [61] |
| Electronic Glass | GB 29495-2013 | Emission standard of air pollutants for electronic glass industry [62] |

The performance of $GTFEE_P$ varies across regions. In general, the eastern region is the most efficient, followed by the central region, and the western region is the least efficient. This finding may be related to the economic structure of each region. The eastern region has a developed economy, and the main industries are advanced manufacturing and service industries. These industries are characterized as high value-added and have less pollutant emissions, so the $GTFEE_P$ of the eastern region is higher than that of the western and central regions. The central region and western region are dominated by manufacturing industries and resource and energy industries, respectively. There are numerous high-energy-consumption and heavy pollution industries, such as steel, chemical and metallurgical industries. Therefore, the $GTFEE_P$ of these two regions is lower than that of the eastern region. Specifically, in 2006, $GTFEE_P$ did not significantly differ among the three regions, with the central and western regions being close in efficiency and slightly higher than the eastern region. Over time, all regions began to show a decreasing trend in $GTFEE_P$, but to different degrees. The eastern, central, and western regions decreased by 14.47%, 34.21%, and 58.56%, respectively. The lowest $GTFEE_P$ values in the eastern and central regions occurred in 2011, while the lowest values in the West occurred in 2014. By 2019, the GTFEE improves to a higher level in both the eastern and central regions. The GTFEE of the western region has improved but still lags behind the other regions. We speculate that the different trends in GTFEE in each region are attributed to the different stages of development in each of the three regions. Although the economies of the eastern and central regions are more developed, the associated cost is massive energy consumption and serious environmental pollution. This situation continued for several years before improving in 2011. The year 2011 marked the beginning of China's 12th Five-Year Plan period, in which China focused more on green development and environmental issues and introduced a series of policies to address air pollution. The western region, on the other hand, is developing at a slower rate, and its energy consumption and air pollutant and $CO_2$ emissions are much smaller in the early years compared to the eastern region, so $GTFEE_P$ was at a higher level. With the rapid economic development, the vast energy consumption and pollutant emissions cause a rapid decline in $GTFEE_P$. However, its economic level is below that of the eastern region, so its level of energy savings and pollution control measures is also relatively lower. This situation leads to a much larger decline in $GTFEE_P$ in the western region than in the eastern region. The efficiency results for different regions are consistent with those of existing studies [54,63,64], where the efficiency levels are highest in the eastern region and lowest in the western region.

This section analyzes the annual average global static $GTFEE_P$ in China, for the whole country and the three study regions. The analysis finds that the change nodes of the national average annual $GTFEE_P$ are closely related to the Five-Year Plan and vary by region.

*3.3. GML Index and Its Decomposition Analysis*

To analyze the dynamics of GTFEE and its variation, we use the GML index to further analyze $GTFEE_P$. The results are shown in Figure 6. During the period 2006–2011, the GML index was low for both the whole country and the study regions, which means that the average annual $GTFEE_P$ in China during that period exhibited a decreasing trend. Decomposing the GML index into EC and TC, we find that EC fluctuates near 1 during

this period, while TC, similar to the GML index, is almost always less than 1. This finding indicates that the decline in GTFEE_P during this period is mainly attributed to technological regression. During the period 2011–2014, the GML index, EC and TC fluctuated above and below 1, and the annual average GTFEE_P in China fluctuated during this period. During the period 2014–2019, the national average annual GTFEE_P started to rise and the GML index and TC were greater than 1, with the exception of a few years. Thus, technological progress was the main reason for the GTFEE_P increase during this period.

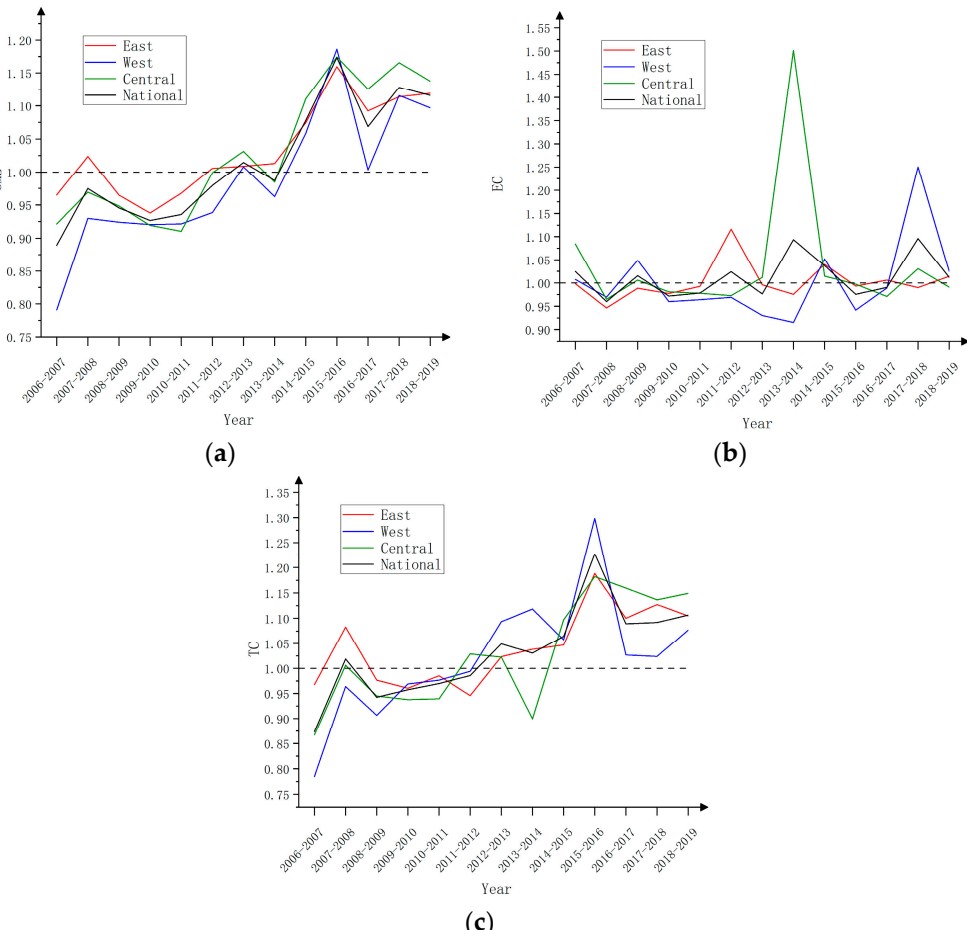

**Figure 6.** National and regional GML indices and their decomposition results: (**a**) GML, (**b**) EC (**c**) TC.

Regionally, GML indices for the eastern and central regions are generally above 1 from 2011 to 2019, which indicates that the GTFEE_P for both the regions basically show an increasing trend since 2011. Further decomposition of the GML index into EC and TC reveals that EC fluctuates above and below 1 in these two regions, while TC is basically higher than 1. This finding indicates that efficiency growth in these two regions is largely dependent on technological progress, rather than the catching up of technological efficiency. The performance of the western region is different from the central and eastern regions, where the GML index has only started to exceed 1 since 2014 (GTFEE exhibits an increasing trend). The efficiency growth in the western region is also largely dependent on technological progress.

We further analyzed the average EC and TC of each province for 14 years and classified them by different regions. The results are shown in Figure 7. TC tends to be higher in the eastern provinces, with half exceeding the national average. Although TC in most of the central provinces exceeds 1 (representing technological progress), these provinces remain below the national average level. TC in the majority of the western provinces is higher than 1, but EC is lower compared to other regions, suggesting that the technical efficiency

improvements in the western provinces are worse and cannot catch up with the efficiency improvements in the eastern and central regions.

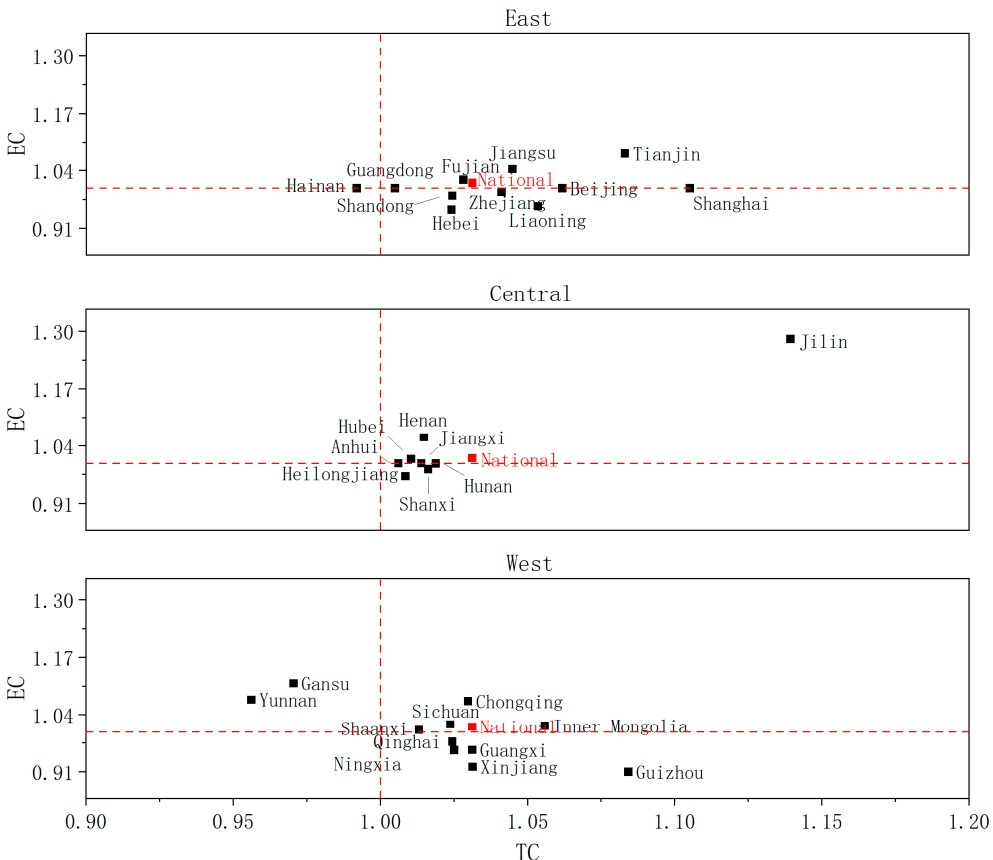

**Figure 7.** Average EC and TC by province over a 14-year period.

This section presents a dynamic analysis of GTFEE$_P$ using the GML index. The results show that the main source of efficiency change is TC, both nationally and regionally, but that the timing of its change turnaround differs. The results are identical to those of existing studies [10,19], where the main driver for improving efficiency was TC.

### 3.4. Analysis of Ranking Changes

The efficiency calculated by DEA is relative efficiency, so in addition to analyzing the efficiency values, we also analyze the changes in the rankings of the provinces. This section ranks the average efficiency of each five-year plan for each province and subtracts the ranking of the beginning period (11th Five-Year Plan) from the ranking of the ending period (13th Five-Year Plan), with a positive number indicating an improved ranking and a negative number representing a decrease in ranking. The results are shown in Table 5.

Among the top ten provinces during the 11th Five-Year Plan period, six provinces were able to maintain their efficiency levels and remained in the top ten until the 13th Five-Year Plan period. Among the five provinces ranked last in the 11th Five-Year Plan period, only Jilin rose to the middle ranking (13th) during the 13th Five-Year Plan period, while the other provinces remained at the bottom. This phenomenon suggests that we should focus on provinces with lower rankings that have difficulty achieving higher efficiency rankings by virtue of their own development conditions.

**Table 5.** Change in Provincial Ranking.

| Province | Area | GTFEE_P Rank | | | |
|---|---|---|---|---|---|
| | | 11th Five-Year Plan | 12th Five-Year Plan | 13th Five-Year Plan | Ranking Changes * |
| Beijing | East | 9 | 2 | 2 | 7 |
| Fujian | East | 8 | 8 | 10 | −2 |
| Guangdong | East | 4 | 1 | 7 | −3 |
| Hainan | East | 1 | 5 | 5 | −4 |
| Hebei | East | 23 | 23 | 26 | −3 |
| Jiangsu | East | 13 | 14 | 12 | 1 |
| Liaoning | East | 25 | 24 | 25 | 0 |
| Shandong | East | 19 | 17 | 20 | −1 |
| Shanghai | East | 21 | 6 | 3 | 18 |
| Tianjin | East | 22 | 19 | 6 | 16 |
| Zhejiang | East | 12 | 9 | 15 | −3 |
| East average | | 14.3 | 11.6 | 11.9 | 2.4 |
| Anhui | Central | 2 | 3 | 1 | 1 |
| Henan | Central | 15 | 11 | 4 | 11 |
| Heilongjiang | Central | 20 | 18 | 21 | −1 |
| Hubei | Central | 10 | 12 | 16 | −6 |
| Hunan | Central | 7 | 7 | 14 | −7 |
| Jilin | Central | 27 | 25 | 13 | 14 |
| Jiangxi | Central | 3 | 4 | 8 | −5 |
| Shanxi | Central | 28 | 28 | 27 | 1 |
| Central average | | 14.0 | 13.5 | 13.0 | 1.0 |
| Gansu | West | 11 | 21 | 22 | −11 |
| Guangxi | West | 5 | 13 | 17 | −12 |
| Guizhou | West | 16 | 22 | 23 | −7 |
| Inner Mongolia | West | 30 | 29 | 24 | 6 |
| Ningxia | West | 29 | 30 | 30 | −1 |
| Qinghai | West | 26 | 26 | 28 | −2 |
| Shaanxi | West | 17 | 20 | 19 | −2 |
| Sichuan | West | 6 | 10 | 11 | −5 |
| Xinjiang | West | 24 | 27 | 29 | −5 |
| Yunnan | West | 14 | 15 | 18 | −4 |
| Chongqing | West | 18 | 16 | 9 | 9 |
| West average | | 17.8 | 20.8 | 20.9 | −3.1 |

* The change is the total change from the 11th Five-Year Plan period to the 13th Five-Year Plan period, a positive number represents a rise in ranking, and a negative number represents a fall in ranking.

Regionally, the changes in the average ranking of the three regions occurred mainly between the 11th Five-Year Plan period and the 12th Five-Year Plan period. This finding shows that provinces that paid attention to air pollution control during the 12th Five-Year Plan period were able to act quickly when the government proposed carbon emission-related policies during the 13th Five-Year Plan period, thus retaining their rankings. In addition, from the change in the average ranking of each region, the average ranking of the eastern region rose 2.4 places, the central region remained almost unchanged, while the western region dropped 3.1 places. This finding confirms our previous statement that in less efficient regions, it is difficult to keep up with the rate of efficiency improvement in the more efficient regions, so the rankings of provinces in these regions tend to fall. This situation is also evidenced by the number of provinces that fell in the rankings among the regions, with approximately half of the number of provinces in the eastern and central regions and 9 of the 11 provinces in the western region falling in the rankings.

This section analyzes the ranking and changes in ranking for each province in the three five-year plans and summarizes them by region. The analysis revealed that the ranking changes mainly during the period from the 11th Five-Year Plan to the 12th Five-Year Plan, while the period from the 12th Five-Year Plan to the 13th Five-Year Plan is relatively stable. The change in ranking varies by region, with more provinces in the west declining in ranking than in the east and central regions. Combined with the analysis of the results in 3.2 and 3.3, the decline in the western ranking is not attributed to lower efficiency but to

the notion that the rate of improvement lower than that of other regions. Therefore, for the western region, more support is needed to prevent serious polarization.

## 4. Conclusions

In the context of pollution and carbon reduction, this paper divides the environmental impacts caused by energy use into two issues, climate change and air pollution, and separately calculates GTFEE using various undesirable output indicators and finds a more robust indicator system among them. Through this index system, we choose the NDDF DEA model as our assessment method to measure China's GTFEE between the 11th and 13th Five-Year Plan periods and use the GML index for dynamic analysis. For the analysis of the results, in addition to the changes in GTFEE, we also compared the changes in the ranking of the provinces in these three five-year plans. The findings of this paper are presented as follows:

1.  When assessing GTFEE, different air pollutant indicators have different inefficiency values. Use of the composite indicator to refer to air pollutants yields more robust results.
2.  During the study period, the national average annual efficiency level was low (highest value of 0.57 and lowest value of 0.32) and showed a decreasing and then increasing trend. The GTFEE showed a decreasing trend during the 11th Five-Year Plan period, although the energy inefficiency values decreased. The environmental (undesirable outputs) inefficiency values, on the other hand, started to decline only during the 12th Five-Year Plan period and 13th Five-Year Plan period.
3.  The change in the GML index varies by region, with the eastern and central regions having the lowest GTFEE in 2011, the western region having the lowest value in 2014, and all regions relying mainly on technological progress for efficiency improvements. Average TC tends to be higher in the eastern provinces. Although the provinces in the central region are mostly experiencing technological progress (average TC higher than 1), they are still not at the national average level. In the provinces in the western region are also mostly experiencing technological progress, their efficiency improvement (EC) rate is not as high as that in the eastern and central provinces, and the GTFEE is low in comparison.
4.  Changes in the ranking of the provinces occurred mainly during the 11th to 12th Five-Year Plan period. The western provinces have not improved as quickly as provinces in the eastern and central regions, resulting in a decline in the ranking of most of the western provinces.

China's GTFEE is closely related to the five-year plan, and energy efficiency policies alone are not sufficient to support an increase in GTFEE; policies regarding atmospheric environmental management are equally important. China has also made practical actions, through the completed 12th Five-Year Plan, the Air Pollution Prevention and Control Action Plan and other policies, which have greatly reduced the air pollutant emissions. Simultaneously, China's proposed carbon peak and carbon neutral targets have slowed the growth of $CO_2$ emissions. With the joint efforts of many parties, improvements in GTFEE have been achieved. In addition, China is a vast country with different stages of development in each region, so it is necessary to formulate different policies accordingly. The eastern region is ahead of the central and western regions in economic development and is relatively more efficient, but its energy consumption and air pollutant and $CO_2$ emissions exceed those of the other two regions. For the eastern region, under the premise of ensuring economic development, more attention should be paid to breakthroughs in energy-saving technology and environmental management technology to improve the energy structure to reduce the negative environmental impact caused by economic development. For the western region, although its GTFEE has improved after the 12th Five-Year Plan, it lags behind the eastern and central regions. Therefore, the western region needs to learn advanced environmental management tools and management methods from the eastern and central regions. The government can also assist when necessary to avoid polarized development.

**Author Contributions:** Conceptualization, Z.Z.; methodology Z.Z.; formal analysis, Z.Z. and W.X.; investigation, Z.Z., resources W.X. and Z.C.; data curation, Z.Z., W.X. and Z.C.; writing—original draft preparation, Z.Z.; writing—review and editing, Z.Z. and W.X.; supervision, Z.Z.; project administration, Z.Z. All authors have read and agreed to the published version of the manuscript.

**Funding:** This research received no external funding.

**Institutional Review Board Statement:** Not applicable.

**Informed Consent Statement:** Not applicable.

**Data Availability Statement:** The data presented in this study are all from the statistical data officially released by China and are explained in Section 2.3.

**Conflicts of Interest:** The authors declare no conflict of interest.

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
