# Peer review of "China’s Green Total Factor Energy Efficiency Assessment Based on Coordinated Reduction in Pollution and Carbon Emission: From the 11th to the 13th Five-Year Plan"

_sustainability, doi:10.3390/su15097301_

Round 1

Reviewer 2 Report

The study takes into account important problems for the development of China under the conditions of assuming the SDGs.

We believe that the work can be improved if the authors bring clarifications and additions, as follows:

- in the summary it is necessary to indicate the added value of the work and not just the simple statement of some results. The 4 results only state what was obtained, they do not indicate the added value, neither from the perspective of knowledge for technological transfer, nor as a method, nor as an approach!

The GTFEE method is used for historical analysis. In this context, what is the motivation for selecting the model compared to others and why is the assessment "is more relevant for energy efficiency assessment because it allows inputs and outputs to be adjusted in different proportions" important in the analysis.

The purpose of the analysis, according to the authors, is "In contrast to the traditional DEA model based on the radial distance function, the NDDF model can make specific suggestions for energy use, help us to explore more targeted energy adjustment strategies in energy management and regulation", but, in the paper, the scientific foundations of the use of the model are not highlighted.

The second aspect, of the use of GTFEE at the regional level, is also without scientific arguments for application. Research methods in economics are versatile, but their usefulness lies in the scientific significance and robustness of the method and not in their "adapted" application to obtain the desired results!

The analysis remains at a superficial level, comparing the "performances" of China's regions, but without taking into account the significant difference given by the economic structure in the fields of activity, which implies different effects on the results (output) and pollution.

Beyond the calculation exercise, at the regional level, using the selected methods, the usefulness for the development of knowledge is not clearly highlighted a) neither in the field of analysis of the adjustments of the desired results depending on the negative externalities of the various forms of pollution and b) nor from the perspective of making the measurement of progress more flexible reducing pollution at the regional level.

We consider that the analysis should be better outlined on the following aspects:

- the novelty of the method and the elements of originality

- the scientific substantiation of the regional approach and the possible clear effects in the field of policies, starting from the reality of the gaps in the structure, performance and economic efficiency at the regional level.

- the usefulness of the results from the perspective of regional policy dynamics

Reviewer 3 Report

Green Total Factor Energy Efficiency (GTFEE) measures are increasing their popularity in the academic literature to balance between environmental concerns and GDP growth. The paper is a good attempt to explore the GTFEE by taking China as case study by examining the long period (2006 to 2019). The paper used robust data and methodology and conclusions are highly relevant for other countries to replicate such studies. I suggest authors to cite ‘Chand, R., Raju, S. S., & Reddy, A. A. (2015). Assessing performance of pulses and competing crops based on market prices and natural resource valuation. Journal of Food Legumes, 28(4), 335-340’ to widen the readership even from other countries by future researchers who are working on similar problems in other countries.  

Round 2

Reviewer 1 Report

I think the authors have carefully checked and revised, and some grammar mistakes should be check.

Author Response

Thank you very much for your suggestions on the paper. We have checked the grammar in the manuscript. We did not find obvious grammatical errors. Therefore the changes marked in red are not in the version we uploaded.

Reviewer 2 Report

No additional comments

Author Response

Thank you very much for your professional advice on the paper.